# Andrographolide Inhibits Biofilm and Virulence in *Listeria monocytogenes* as a Quorum-Sensing Inhibitor

**DOI:** 10.3390/molecules27103234

**Published:** 2022-05-18

**Authors:** Tao Yu, Xiaojie Jiang, Xiaobo Xu, Congyi Jiang, Rui Kang, Xiaobing Jiang

**Affiliations:** 1School of Life Sciences & Basic Medicine, Xinxiang University, Xinxiang 453000, China; yutao@xxu.edu.cn (T.Y.); jiangxiaojie1987@hotmail.com (X.J.); xuxb0423@163.com (X.X.); 2Key Laboratory of Biomedicine and Health Risk Warning of Xinxiang City, Xinxiang University, Xinxiang 453000, China; 3Henan Engineering Laboratory for Bioconversion Technology of Functional Microbes, College of Life Sciences, Henan Normal University, Xinxiang 453007, China; 15225963923@163.com (C.J.); K18637233560@163.com (R.K.)

**Keywords:** *Listeria monocytogenes*, andrographolide, quorum sensing, biofilm, virulence

## Abstract

*Listeria monocytogenes* is a major foodborne pathogen that can cause listeriosis in humans and animals. Andrographolide is known as a natural antibiotic and exhibits good antibacterial activity. We aimed to investigate the effect of andrographolide on two quorum-sensing (QS) systems, LuxS/AI-2 and Agr/AIP of *L. monocytogenes*, as well as QS-controlled phenotypes in this study. Our results showed that neither *luxS* expression nor AI-2 production was affected by andrographolide. Nevertheless, andrographolide significantly reduced the expression levels of the *agr* genes and the activity of the *agr* promoter P_2_. Results from the crystal violet staining method, confocal laser scanning microscopy (CLSM), and field emission scanning electron microscopy (FE-SEM) demonstrated that andrographolide remarkably inhibited the biofilm-forming ability of *L. monocytogenes* 10403S. The preformed biofilms were eradicated when exposed to andrographolide, and reduced surviving cells were also observed in treated biofilms. *L. monocytogenes* treated with andrographolide exhibited decreased ability to secrete LLO and adhere to and invade Caco-2 cells. Therefore, andrographolide is a potential QS inhibitor by targeting the Agr QS system to reduce biofilm formation and virulence of *L. monocytogenes*.

## 1. Introduction

*Listeria monocytogenes*, a Gram-positive bacterium, is recognized as an important foodborne pathogen that can cause listeriosis, a severe invasion infection in humans and animals [1]. *L. monocytogenes* appears to be widely distributed in nature and can easily enter food processing environments. The ability of *L. monocytogenes* to persist for long periods in equipment and environments of the food industry increases the risk of contaminated food and outbreaks of listeriosis. The typical clinical symptoms of listeriosis include septicemia, meningitis, abortion, and neonatal death [2]. Listeriosis is a rare but serious disease with high case-fatality rates [3]. Food contaminated by *L. monocytogenes* is recognized as the main vehicle for acquiring listerial infections [4]. Therefore, controlling *L. monocytogenes* in the food industry is an important issue for food safety and public health. 

Quorum sensing (QS) is a cell-to-cell communication process that allows bacteria to behave coordinately through the exchange of extracellular signaling molecules called autoinducers [5]. Once the concentration of autoinducer reaches a certain threshold, QS is activated and regulates the expression of a specific set of genes [5]. Therefore, QS is also an important regulatory mechanism in bacteria, and it controls a variety of physiological activities such as bioluminescence, biofilm formation, and virulence [6,7,8].

In *L. monocytogenes*, two QS systems, LuxS/autoinducer 2 (AI-2) and Agr/autoinducing peptide (AIP), have been identified. The LuxS/AI-2 appears in many species of Gram-negative and Gram-positive bacteria [9]. LuxS is the key enzyme in the AI-2 biosynthesis pathway, and it is highly conserved in bacteria [10]. Thus, AI-2 is considered a QS signaling molecule for interspecies communication. In *L. monocytogenes*, LuxS is involved in the repression of biofilm formation [11,12]. The Agr/AIP consists of the four-gene operon *agrBDCA,* whose expression is driven by the P_2_ promoter upstream of *agrB* [13]. Many studies have suggested that the Agr system contributes to biofilm formation and virulence in *L. monocytogenes* [13,14,15]. As QS is vital in the regulation of many important phenotypes of bacteria, interfering with the QS networks is considered to be a potential way to prevent food contamination and human infection. Although several synthetic and natural compounds, such as 4-hydroxy-2,5-dimethyl-3(2H)-furanone, ambuic acid, and lactocin AL705, have been found to be capable of inhibiting the QS systems in *L. monocytogenes* [16,17,18], it is still urgent to search for more novel QS inhibitors.

*Andrographis paniculata* (Burm. F) Nees is a well-known traditional medicinal and edible plant in China. Andrographolide (its chemical structure is presented in Figure 1A), a diterpene lactone, is the major bioactive component of *A. paniculata*. Andrographolide has attracted much attention because of its multiple medicinal properties, such as antibacterial and anti-inflammatory activities [19]. Recently, several studies have demonstrated that andrographolide can inhibit the virulence and pathogenicity of *Pseudomonas aeruginasa* and *Escherichia coli* by interfering with the bacterial QS system [20,21]. However, its effect on the QS systems of *L. monocytogenes* is still unknown. 

Therefore, the aim of this study was to investigate the QS inhibitory activity of andrographolide against *L. monocytogenes*. The effects of andrographolide on biofilm formation and mature biofilms were evaluated by identifying changes in biofilm biomass, morphology, and architecture. Swimming and swarming motilities of *L. monocytogenes* were measured in the presence of andrographolide. In addition, the effects of andrographolide treatment on the secretion of hemolysin, as well as on the ability to adhere to and invade Caco-2 cells, were also investigated to determine its anti-infective activity.

## 2. Results

### 2.1. Determination of Minimum Inhibitory Concentration (MIC) of Andrographolide

In this study, the antimicrobial susceptibility of *L. monocytogenes* to andrographolide was assessed, and the concentration range of andrographolide was 0.25–3 mg/mL with 0.5 mg/mL steps. The MIC of andrographolide against *L. monocytogenes* 10403S was 2 mg/mL.

### 2.2. Growth in the Presence of Andrographolide

Growth curves of *L. monocytogenes* 10403S in the presence of sub-MICs of andrographolide are shown in Figure 1B. Our results showed that andrographolide at concentrations of 1 mg/mL or below had no inhibitory effect on bacterial growth compared with the control.

### 2.3. Effect of Andrographolide on luxS Expression and AI-2 Production

As shown in Figure 2A, the presence of andrographolide did not result in a significant change in the expression level of *luxS*. Results from the bioluminescence assay confirmed that *L. monocytogenes* 10403S had the ability to produce AI-2 (Figure 2B). However, the addition of sub-MICs of andrographolide did not significantly affect AI-2 production (Figure 2B).

### 2.4. Effect of Andrographolide on agr Expression and the agr Promoter (P_2_) Activity

The expression levels of *agrBD*, *agrC*, and *agrA* genes were significantly decreased (*p* < 0.05) in *L. monocytogenes* 10403S after exposure to andrographolide (Figure 2A). The P_2_ activity was determined by the *β*-galactosidase assay in this study. As shown in Figure 2C, andrographolide treatment at 0.125, 0.25, 0.5, and 1 mg/mL reduced the P_2_ activity by approximately 26%, 29%, 36%, and 46%, compared with the control. These data demonstrated that andrographolide possesses anti-QS capacity by inhibiting the Agr/AIP system in *L. monocytogenes*. 

### 2.5. Inhibition of Biofilm Formation by Andrographolide 

Treatment with andrographolide at concentrations of 0.125, 0.25, 0.5, and 1 mg/mL significantly reduced (*p* < 0.05) the biofilm biomass by 17.5%, 28.3%, 57.7%, and 73.7%, respectively (Figure 3A). The difference in biofilm-forming ability is not due to a difference in growth because the addition of andrographolide had no effect on the viability of planktonic cells compared with that of the control (Figure 3B). 

Confocal laser scanning microscopy (CLSM) and field emission scanning electron microscopy (FE-SEM) were applied to observe the morphology of *L. monocytogenes* biofilms. CLSM images showed a compact biofilm structure in the control (Figure 3C–E). In contrast, reduced thickness of biofilms was observed following andrographolide treatment (Figure 3C–E). Similar results were also obtained with the FE-SEM images, in which biofilms showed a more scattered appearance in the presence of andrographolide (Figure 3F).

### 2.6. Effect of Andrographolide on Preformed Biofilms

In the present study, andrographolide at sub-MICs (0.5 and 1 mg/mL) and inhibitory concentrations (2 and 4 mg/mL) was used to investigate the effect of andrographolide on mature biofilms. As shown in Figure 4A, andrographolide at 0.5 mg/mL (1/4MIC) resulted in only a minor reduction (4.6%) in the preformed biofilms (*p* > 0.05). When treated with andrographolide at 1 (1/2MIC), 2 (MIC), and 4 mg/mL (2MIC), mature biofilms were eradicated by 13.2%, 24.3%, and 50.2%, respectively (*p* < 0.05). The counting results of surviving cells in treated biofilm (Figure 4B) further confirmed the eradication effect of andrographolide at concentrations above 1/2MIC on preformed biofilms of *L. monocytogenes* 10403S. Our results also showed that the viability of planktonic cells in the culture supernatant was not influenced by andrographolide (0.5 to 4 mg/mL; Figure 4C).

In addition to quantitative analysis, treated biofilms were also visualized using CLSM and FE-SEM. CLSM images showed complete and uniformly distributed biofilm in the control experiment, whereas andrographolide treatment, especially at 2 and 4 mg/mL, significantly reduced the number of attached cells (Figure 4D–F). The FE-SEM images also indicated decreased adherence in the andrographolide-treated biofilms (Figure 4G). 

### 2.7. Effect of Andrographolide on Motility

In this study, swimming and swarming motilities of *L. monocytogenes* 10403S grown with andrographolide (0.125, 0.25, 0.5, and 1 mg/mL) were evaluated. After incubation at 25 °C, no significant difference (*p* > 0.05) in the mean diameter of the swim and swarm ring was observed between the andrographolide treatment group and the control (Appendix A). Expression levels of the flagella gene (*flaA*) and motility-related genes (*motA* and *motB*) were not affected by andrographolide (Appendix A). These data suggest that andrographolide has no effect on swimming and swarming motilities of *L. monocytogenes*.

### 2.8. Effect of Andrographolide on Virulence Factors

As shown in Figure 5A, the hemolytic activity of *L. monocytogenes* 10403S was significantly reduced (*p* < 0.05) by the addition of andrographolide. When treated with andrographolide at 0.03125, 0.0625, 0.125, and 0.25 mg/mL, the hemolysis rate of 10403S was 32.2%, 26.6%, 15.4%, and 4.0%, respectively. 

The capacity of *L. monocytogens* 10403S to attach and invade Caco-2 cells was also determined in the presence or absence of andrographolide. When treated with andrographolide at 0.125, 0.25, 0.5, and 1 mg/mL, adhesion of 10403S was decreased to 59.4%, 53.3%, 48.5%, and 41.6%, respectively (Figure 5B); the invasion rate was reduced to 83.8%, 76.0%, 70.1%, and 67.7%, respectively (Figure 5C). 

Expression levels of virulence-associated genes (*prfA*, *plcA*, *hly*, *mpl*, *actA*, *plcB*, *inlA*, and *inlB*) were investigated in this study. All virulence genes tested, except *mpl*, were down-regulated (*p* < 0.05) after treatment of andrographolide (Figure 5D).

## 3. Discussion

QS regulates many important physiological processes in several pathogenic bacteria, such as biofilm formation and the production of virulence factors. *L. monocytogenes* has been considered an important foodborne pathogen all over the world, and it has the ability to form biofilms in food-processing environments [22,23]. *L. monocytogenes* growing in biofilms is significantly more resistant to disinfection than its free-living counterparts, leading to the increased risk of food contamination by this bacterium. Consumption of *L. monocytogenes*-contaminated food may cause listeriosis, which threatens public health. Therefore, controlling biofilm formation and virulence of *L. monocytogenes* is one of the most important issues in the food industry. Application of QS inhibitor to decrease biofilm formation and virulence by blocking QS is considered to be an effective risk-reducing way of food contamination by *L. monocytogenes*. 

Andrographolide exhibits not only excellent antibacterial activity but also inhibitory effects on QS. Guo et al. (2014) have reported that andrographolide interferes with the LuxS/AI-2 QS system in avian pathogenic *E. coli* and inhibits the AI-2 activity [21]. *P. aeruginosa* has two acylated homoserine lactone (AHL)-based QS systems, LasI/LasR and RhlI/RhlR. It has been reported that 14-alpha-lipoyl andrographolide, a derivative of andrographolide, represses the expression levels of QS genes and AHL production, and consequently, biofilm formation regulated by QS systems is inhibited in *P. aeruginosa* PAO1 [20].

In the present study, the effects of andrographolide on two QS systems were investigated in Gram-positive bacterium *L. monocytogenes*. Unlike *E. coli*, the LuxS/AI-2 system of *L. monocytogenes* was not affected by andrographolide. Neither *luxS* expression nor AI-2 production changed after exposure to this agent. In *E. coli*, LsrB, the receptor for signaling molecule AI-2, has been identified; however, a similar receptor has not been reported in *L. monocytogenes* [24]. Some scholars support the view that AI-2 plays the role of metabolite rather than a QS signaling molecule in *L. monocytogenes* [11,12]. Thus, it is not surprising that andrographolide exhibited different effects on the LuxS/AI-2 systems in *E. coli* and *L. monocytogenes*. Interestingly, the presence of andrographolide significantly reduced the expression levels of the *agr* genes and the activity of the *agr* promoter P_2_, indicating that andrographolide is a potential QS inhibitor targeting the Agr/AIP system in *L. monocytogenes*. To our knowledge, the Agr/AIP system is a typical QS system in Gram-positive bacteria. In *L. monocytogenes*, AgrD, a precursor peptide, is processed into an active signaling molecule AIP by AgrB, and then, AIP is released outside the cells. When the concentration of AIP in the extracellular space reaches a certain threshold, it activates the two-component system consisting of AgrC (histidine kinase) and AgrA (response regulator) by binding specifically to AgrC. AgrA not only regulates the expression of target genes but also autoregulates the *agr* operon via driving the P_2_ promoter upstream of *agrB*. So how does andrographolide interfere with the Agr/AIP system? Is it by inhibiting AIP biosynthesis, by disrupting the interaction between AIP and AgrC, or by other means? Further studies are needed to clarify the mechanism of andrographolide in the Agr QS system of *L. monocytogenes*.

We tried to investigate the effect of andrographolide on AIP production by quantifying actual AIP concentrations in bacterial culture supernatants using high-performance liquid chromatography (HPLC) but failed. The interference of the complex components and the high levels of peptides in the BHI medium may be one of the reasons [25]. It is predicted that peptide-signaling molecules are more metabolically costly, and their concentrations in bacterial culture were much lower than other types of signaling molecules [26]. Therefore, there is also the possibility that *L. monocytogenes* produce a very small amount of AIP, which also makes it difficult to quantify AIP by HPLC. Since andrographolide can block the Agr system, the phenotype controlled by this system may also be affected by andrographolide. Previous studies have confirmed positive regulation by the Agr system on biofilm formation in *L. monocytogenes* [14]. Results from the crystal violet staining method, CLSM, and FE-SEM demonstrated that the addition of andrographolide remarkably inhibited the biofilm-forming ability of *L. monocytogenes* 10403S. The mature biofilms formed by *L. monocytogenes* are difficult to remove even after routine cleaning and disinfection procedures. Therefore, the effects of andrographolide on preformed biofilms were also evaluated in this study. After exposure to andrographolide, decreased biofilm biomass and surviving cells were observed in treated biofilms. Andrographolide could not only destroy the structure of mature biofilms but also reduce the number of bacterial cells in biofilms. 

Flagellum-mediated motilities, swimming motility, and swarming motility play a significant role in the first stages of biofilm formation [27]. It has been reported that these flagellar motility genes (*flaA*, *motA*, and *motB*) play important roles in the biofilm formation of *L. monocytogenes* [28,29]. In the current study, swimming and swarming motilities and expression of flagellar motility genes were not influenced by andrographolide.

*L. monocytogenes* is an intracellular pathogen that can cause severe invasive infections in humans. In fact, the process of *L. monocytogenes* infection of host cells is regulated by virulence factors [30]. Given that the Agr system is associated with virulence and invasion of *L. monocytogenes*, effects of andrographolide on some key virulence factors were also investigated in the current study. Listeriolysin O (LLO) encoded by the *hly* gene is a Listeria-specific hemolysin [31]. Our results showed that andrographolide significantly reduced the hemolytic activity of LLO. Notably, the LLO activity was almost completely inhibited in the presence of andrographolide at 0.25 mg/mL (1/8MIC), suggesting the excellent inhibitory effect of andrographolide at low concentration on the LLO activity. As a result, concentrations below 0.25 mg/mL were selected to evaluate the inhibitory effect of andrographolide on the LLO activity. In addition, the ability of *L. monocytogenes* to adhere to and invade Caco-2 cells was also suppressed when exposed to andrographolide at concentrations ranging from 0.125 to 1 mg/mL. Consistently, most virulence genes tested in our study were down-regulated after treatment of andrographolide.

Andrographolide is considered a potential QS inhibitor for some Gram-negative bacteria. The effect of andrographolide on QS systems of *L. monocytogenes* was investigated in our study. Andrographolide interfered with the Agr QS system of *L. monocytogenes* and consequently inhibited bacterial behaviors regulated by this QS system. *L. monocytogenes* treated with andrographolide exhibited decreased ability to form biofilms, secrete LLO, and adhere to and invade Caco-2 cells. Therefore, andrographolide has the potential to control *L. monocytogenes* contamination in the food industry as a QS inhibitor. 

## 4. Materials and Methods

### 4.1. Bacterial Strains and Growth Conditions

Wild-type *L. monocytogenes* 10403S was obtained as a gift from Prof. Qin Luo and was grown in brain heart infusion (BHI; Oxoid Ltd., Basingstoke, UK) broth at 37 °C. *E. coli* DH5α was purchased from Bomaide Gene Technology Co., Ltd. (Beijing, China) and was grown in Luria-Bertani (Huankai Ltd., Guangzhou, China) broth. *Vibrio harveyi* BB170, an AI-2 reporter strain, was kindly provided by Xiangan Han (Shanghai Veterinary Research Institute) and was grown in AB medium at 30 °C. 

### 4.2. Determination of MIC of Andrographolide

Andrographolide (purity > 98%, CAS 5508-58-7) was purchased from Shanghai Macklin Biochemical Co., Ltd. (Shanghai, China). Stock solutions were prepared by dissolving andrographolide in dimethyl sulfoxide (DMSO, Macklin, Shanghai, China). The final concentration of DMSO in all samples was 0.1%, which had no significant effect on the growth of *L. monocytogenes* 10403S (data not shown). The MIC of andrographolide against *L. monocytogenes* 10403S was determined using the agar dilution method, as described previously [32]. Bacterial cultures were diluted using sterilized saline solution to approximately 10^7^ CFU/mL. One microliter of suspension was inoculated on Mueller–Hinton agar (MHA; Huankai, Guangzhou, China). MHA containing DMSO was used as the blank control. MHA containing DMSO inoculated with bacterial culture was used as the negative control. The MIC was defined as the lowest concentration of andrographolide that prevented the visible growth of *L. monocytogenes*.

### 4.3. Growth Curve Analysis

For growth measurement, *L. monocytogenes* 10403S was inoculated into BHI broth in the absence or presence of andrographolide (0.125, 0.25, 0.5, and 1 mg/mL). BHI with DMSO served as the negative control. The samples were incubated in a Bioscreen C microbiology reader (Growth Curves, Helsinki, Finland) at 37 °C for 24 h, with *OD*_600nm_ measurements collected hourly and used to generate bacterial growth curves. 

### 4.4. Gene Expression Analyses

Quantitative real-time PCR (qRT-PCR) was performed to measure the relative expression levels of the target genes using the LightCycler 96 real-Time PCR system (Roche, Basel, Switzerland), as described previously [33]. Primers for eight virulence genes are listed in Appendix A, and other primers have been reported in our previous study [33]. To assess the effect of andrographolide on gene expression, *L. monocytogenes* 10403S was grown in BHI supplemented with or without andrographolide (1 mg/mL). 16S rRNA was used as a reference gene, and the fold changes of the target genes were calculated using the 2^−ΔΔCt^ method. 

### 4.5. Detection of AI-2

The bioluminescence assay was used to detect AI-2 as described previously [34] using an EnSpire Multimode Plate Reader (PerkinElmer, Waltham, MA, USA). To investigate the effect of andrographolide on AI-2 production in *L. monocytogenes* 10403S, bacteria were incubated in BHI broth with different concentrations of andrographolide (0.125, 0.25, 0.5, and 1 mg/mL), and the medium containing DMSO was used as the blank control. Results were normalized to the DMSO control.

### 4.6. Determination of the agr Promoter P_2_ Activity

The activity of the *agr* promoter P_2_ was determined in this study. The *agr* promoter (P_2_)-*lacZ* fusion was constructed as described previously [35,36]. *β*-galactosidase activity assay was performed based on the method by Miller [37]. 

### 4.7. Effect of Andrographolide on Biofilm Formation by L. monocytogenes

Biofilms of *L. monocytogenes* were measured using the microplate assay with crystal violet staining in 96-well polystyrene plates (Costar 3599; Corning Inc., Kennebunk, ME, USA), as described previously [38]. Briefly, overnight cultures of *L. monocytogenes* were diluted at 1:100 in fresh BHI broth, and andrographolide was added to the cultures to achieve final concentrations of 0, 0.125, 0.25, 0.5, and 1 mg/mL. BHI with DMSO served as the blank control. Two hundred microliters of bacterial cultures were transferred into wells of the microtiter plate. The plates were incubated at 37 °C for 48 h without agitation. To determine the growth of planktonic cells, 100 μL of cultures were centrifuged, and the pellets were resuspended in 1 mL of sterile saline. The bacterial cultures were 10-fold serially diluted, and 100 μL volumes were taken for colony counting. To assess biofilm formation, the medium containing planktonic cells was removed after incubation. Then, biofilms were stained with 1% crystal violet solution for 45 min and then decolorized with 95% ethanol. The absorbance at *OD*_595nm_ was measured to determine biofilm production. 

### 4.8. Effect of Andrographolide on Preformed Biofilms by L. monocytogenes

*L. monocytogenes* biofilms were incubated as described above. After 48 h of incubation at 37 °C, the medium was removed, and wells were washed three times with sterile water. Then, fresh BHI broth supplemented with different concentrations of andrographolide was added to the wells, with DMSO serving as the control. Plates were incubated at 37 °C for 4 h. Planktonic cells were quantified as described above. To quantify biofilm biomass, biofilms were stained with crystal violet after removing the medium and then decolorized by 95% ethanol. The absorbance at *OD*_595nm_ was measured. To quantify biofilm bacteria, the wells were washed several times with sterile water. Then, 200 μL of sterile saline was added to each well, and its surface was thoroughly scratched with a plastic pipette tip. Recovered biomasses were vortexed, serially diluted, and plated onto BHI agar. These plates were incubated at 37 °C for 24 h prior to enumeration by counting colonies. 

### 4.9. CLSM Biofilm Formation Assay

Biofilm assay by CLSM was performed as described previously [33]. Biofilms were stained using the Live/Dead BacLight Bacterial viability kit (Molecular Probes, Eugene, OR, USA) and then observed by a Leica TCS-SP8 Confocal Laser Scanning Microscope (Leica-Microsystems, Wetzlar, Germany). To construct the three-dimensional projections of biofilms, the IMARIS 7.1 software (Bitplane, Zürich, Switzerland) was applied to process the CLSM acquisitions. Biofilm biomass and thickness were quantified by the COMSTAT software [39].

### 4.10. FE-SEM Analysis

FE-SEM (Hitachi SU8010; Hitachi, Tokyo, Japan) was applied to observe biofilms of *L. monocytogenes* as described previously [40]. After 48 h of incubation at 37 °C, *L. monocytogenes* biofilms were fixed in 4% glutaraldehyde and then washed with phosphate-buffered saline (PBS; pH 7.4). Samples were progressively dehydrated by passage through a graded series of ethanol solutions from 30% to 100%. The samples were coated with gold in an ion sputter coater (SBC-12; KYKY Technology Co. Ltd., Beijing, China) and observed with FE-SEM.

### 4.11. Motility

Swimming and swarming motilities of *L. monocytogenes* were tested in our study. Briefly, 1 μL of overnight *L. monocytogenes* 10403S culture was inoculated at the center of 0.3% and 0.5% tryptic soy broth (TSB) agar plate with different concentrations of andrographolide, respectively. Plates were cultivated at 25 °C, and the diameter of the bacterial swim and swarm was measured 48 h later.

### 4.12. Effect of Andrographolide on L. monocytogenes Key Virulence Factors

The hemolytic activity of *L. monocytogenes* was assayed as described previously [41]. The bacterial cultures were centrifuged (5500× *g*, 4 °C, 10 min), and 250 μL of supernatant was mixed with 900 μL of hemolysin buffer and 100 μL of sheep red blood cells. BHI broth and the sheep red blood cells treated with 1% Triton X-100 served as the negative control and positive control, respectively. The samples were determined by measuring the absorbance at *OD*_543nm_. The relative hemolysis was presented as a percentage of the positive control.

Adhesion and invasion assays were carried out as previously described [14]. Briefly, bacterial cultures of *L. monocytogenes* were mixed with Caco-2 cells at a multiplicity of infection (MOI) of 100 and then incubated at 37 °C for 1 h. For adhesion assay, cells were washed with pre-warmed PBS and then lysed with ice-cold distilled water. For invasion assay, cells were incubated in DMEM medium with 10 μg/mL gentamicin after washing once with pre-warmed PBS. The cells were washed and lysed according to the steps described above. The lysed cells were plated on BHI agar and incubated at 37 °C for 24 h.

### 4.13. Statistical Analysis

Statistical analysis was performed using the SPSS version 20.0 (SPSS Inc., Chicago, IL, USA). Differences were defined as significant at *p* < 0.05.

## 5. Conclusions

In summary, andrographolide reduced the expression levels of the *agr* genes and the activity of the *agr* promoter P_2_, suggesting the inhibitory effect of andrographolide on the Agr QS system. Andrographolide could not only inhibit biofilm formation but also remove mature biofilms of *L. monocytogenes*, indicating the anti-biofilm activity of andrographolide against *L. monocytogenes*. Additionally, the hemolytic activity and the ability to adhere to and invade Caco-2 cells of *L. monocytogenes* were suppressed by andrographolide. In *L. monocytogenes*, the Agr QS system is involved in biofilm formation and virulence. Thus, andrographolide is a potential QS inhibitor by targeting the Agr QS system to reduce biofilms and virulence of *L. monocytogenes*. 

## Figures and Tables

**Figure 1 molecules-27-03234-f001:**
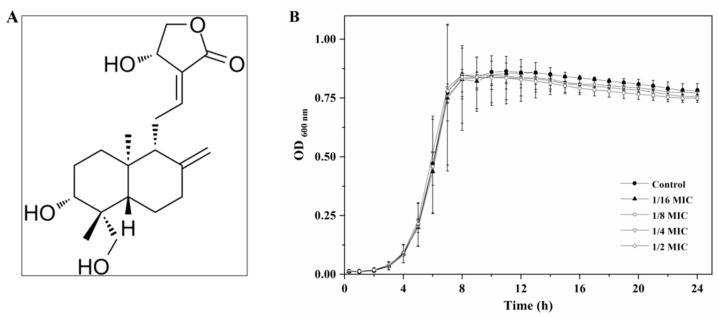
(**A**) Chemical structure of andrographolide and (**B**) growth curves of *L. monocytogenes* in BHI broth with different concentrations of andrographolide.

**Figure 2 molecules-27-03234-f002:**
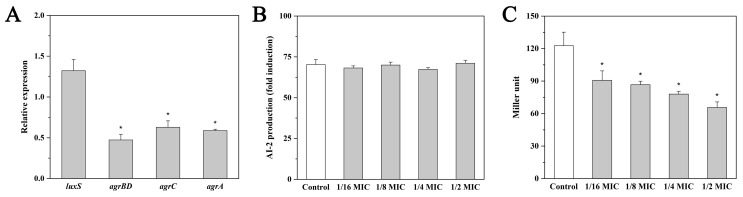
Effect of andrographolide on two QS systems in *L. monocytogenes*. (**A**) Relative expression levels of QS-related genes in *L. monocytogenes* 10403S grown in BHI with or without andrographolide (1 mg/mL). Results are presented as fold changes relative to the expression level of the target gene in *L. monocytogenes* 10403S in BHI. (**B**) The production of AI-2. *Vibrio harveyi* BB170 and *Escherichia coli* DH5α were used as the positive control and the negative control, respectively. The relative activity of AI-2 was presented as a percentage of the positive control. (**C**) The *agr* promoter (P_2_) activity. Error bars represent the standard deviation of triplicate experiments (*n* = 3). The asterisk indicates a value statistically different from that of the control, with *p* < 0.05.

**Figure 3 molecules-27-03234-f003:**
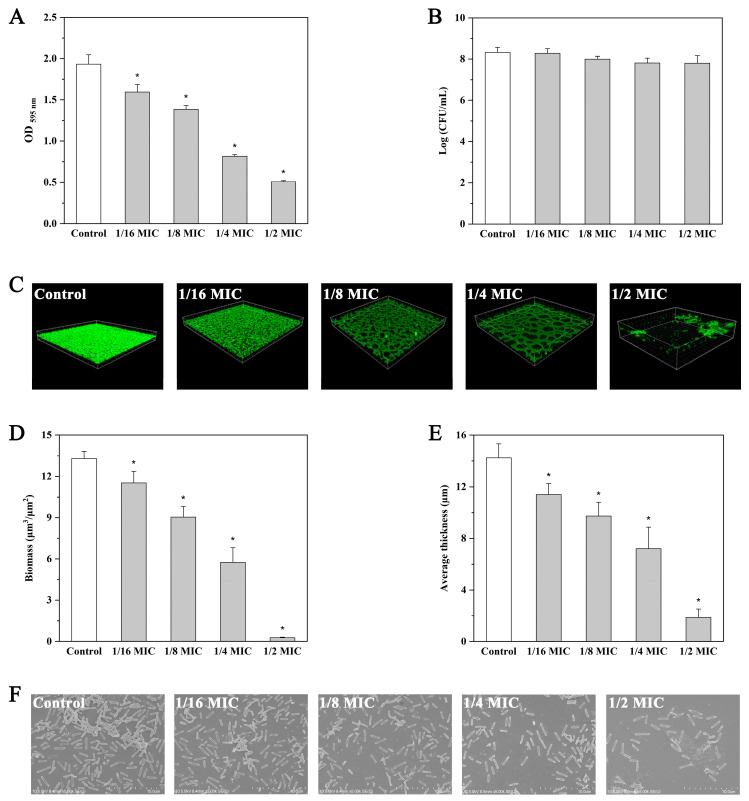
Effect of andrographolide on biofilm formation. (**A**) Biofilm assay by microtiter plater with crystal violet staining. (**B**) Surviving planktonic cells in the bacterial culture. (**C**) CLSM images. (**D**) The biofilm biomass of CLSM analysis. (**E**) Mean thickness of biofilm by CLSM analysis. (**F**) SEM images. Error bars represent the standard deviation of triplicate experiments (*n* = 3). The asterisk indicates a value statistically different from that of the control, with *p* < 0.05.

**Figure 4 molecules-27-03234-f004:**
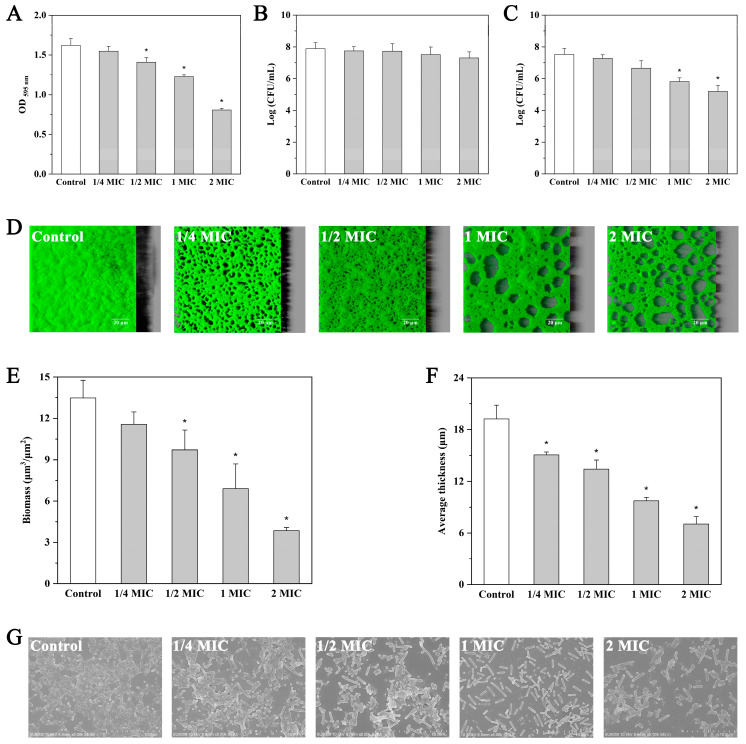
Effect of andrographolide on preformed biofilms. (**A**) Biofilm assay by microtiter plater with crystal violet staining. (**B**) Surviving cells in biofilms. (**C**) Surviving planktonic cells in the bacterial culture. (**D**) CLSM images. (**E**) The biofilm biomass of CLSM analysis. (**F**) Mean thickness of biofilm by CLSM analysis. (**G**) SEM images. Error bars represent the standard deviation of triplicate experiments (*n* = 3). The asterisk indicates a value statistically different from that of the control, with *p* < 0.05.

**Figure 5 molecules-27-03234-f005:**
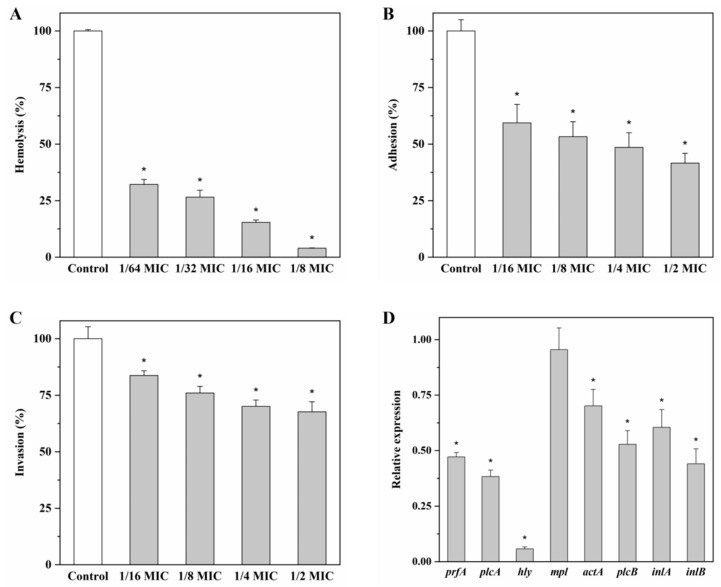
Effect of andrographolide on (**A**) the hemolytic activity, (**B**) the capacity to attach Caco-2 cells, (**C**) the capacity to invade Caco-2 cells, and (**D**) the expression levels of virulence genes, and the results from qRT-PCR are presented as fold changes relative to the expression level of the target gene in *L. monocytogenes* 10403S in BHI. Error bars represent the standard deviation of triplicate experiments (*n* = 3). The asterisk indicates a value statistically different from that of the control, with *p* < 0.05.

## Data Availability

Not applicable.

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
