# Peer review of "Andrographolide Inhibits Biofilm and Virulence in Listeria monocytogenes as a Quorum-Sensing Inhibitor"

_molecules, 2022, doi:10.3390/molecules27103234_

Round 1

Reviewer 1 Report

The manuscript by Yu et al. reports the biological evaluation of andrographolide on two quorum sensing (QS) systems and virulence of L. monocytogenes. In this work, the authors determined the minimum inhibitory concentration (MIC) of andrographolide, and then carried out various assays at sub-MIC levels. Andrographolide was identified to inhibit the Agr/AIP QS system of L. monocytogenes but not the LuxS/AI-2 QS system. It demonstrated its ability to inhibit the formation of L. monocytogenes biofilm significantly at sub-MIC levels and disrupt pre-established L. monocytogenes biofilms at 2x of its MIC. Moreover, it minimised the haemolytic activity of L. monocytogenes and significantly reduced the invasion rate of L. monocytogenes on Caco-2 cells.

The scope of the manuscript is limited as only one compound was included in the study. However, as an extensive biological evaluation has been carried out on that compound, this manuscript can still be considered for publication after all issues have been satisfactorily addressed:

  1. In section 2.6, the biofilm disrupting property of andrographolide was tested at 1/4, 1/2, 1 and 2x of its MIC. However, these concentrations do not match with those labelled in Figure 4.

  1. Some essential information is missing from section 4.2 of the manuscript. What was the concentration of the bacteria used in the assay? What was the final concentration of DMSO in each sample in the assay? Was a solvent control and/or negative control used in the assay? Was a positive control (antibacterial agent with known MIC against the bacteria) used in the assay? If not, why not?

  1. The reasons of the failure of using HPLC to quantify the AIP concentrations in bacterial culture supernatants should be mentioned in the manuscript.

  1. Could the authors please comment (with references) on the reliability of using the activity of agr promoter P2 to estimate the AIP production?

  1. Was a solvent control used in the biofilm formation assay in section 4.7?

  1. In line 307, the method of colony counting should be mentioned in the manuscript.

  1. Sections 4.10 and 4.12: Although the methodology has been previously described and the corresponding references have been provided in the manuscript, the author should still mention the methodology briefly (including essential information of the methods) in the manuscript

  1. I recommend moving Table A1 to the supplementary materials file.

Author Response

1. In section 2.6, the biofilm disrupting property of andrographolide was tested at 1/4, 1/2, 1 and 2x of its MIC. However, these concentrations do not match with those labeled in Figure 4.

Response: Thanks for the reviewer's kind suggestion. Concentrations of andrographolide labeled in Figure 4 were wrong. This is our fault and we are sorry about this. We have provided the correct Figure 4 in revised manuscript.

2. Some essential information is missing from section 4.2 of the manuscript. What was the concentration of the bacteria used in the assay? What was the final concentration of DMSO in each sample in the assay? Was a solvent control and/or negative control used in the assay? Was a positive control (antibacterial agent with known MIC against the bacteria) used in the assay? If not, why not?

Response: Thanks for the reviewer's kind suggestion. Bacterial cultures were diluted using sterilized saline solution to approximately 107 CFU/ml. The final concentration of DMSO in all samples was 0.1%, which had no significant effect on the growth of L. monocytogenes 10403S. MHA containing DMSO was used as the blank control. MHA containing DMSO inoculated with bacterial culture was used as the negative control. The positive control was not used in our study. Several studies on determination of MICs of andrographolide against bacterial strains (such as Streptococcus suis and Staphylococcus aureus) have been reported. Unfortunately, we cannot obtain these strains even if many efforts were made. We are sorry about this and hope to get the reviewer’s understanding. We have added the essential information of section 4.2 in revised manuscript. (Line 286-293)

3. The reasons of the failure of using HPLC to quantify the AIP concentrations in bacterial culture supernatants should be mentioned in the manuscript.

Response: The reviewer’s comment is quite reasonable. We have mentioned the possible reasons of the failure of using HPLC to quantify the AIP concentrations in bacteria culture supernatants in revised manuscript. (Line 229-237)

4. Could the authors please comment (with references) on the reliability of using the activity of agr promoter P2 to estimate the AIP production?

Response: Thanks for the reviewer's kind suggestion. We were planning to quantify the AIP concentration in bacterial culture supernatants using HPLC, but we failed. Since the putative promoter of the agr operon, the P2 promoter has been identified upstream of agrB in L. monocytogenes, we speculate that increased P2 promoter activity results in increased transcriptional level of agrBD, increased expression level of AgrBD and higher production of AIP. Thus, the P2 promoter activity could be determined to indirectly reflect AIP levels. Strictly speaking, the promoter regulates the transcription of gene and the promoter activity reflects the transcription level of genes. In some cases, the transcriptional level of genes is not consistent with the expression level of protein encoded by the genes. At present, we cannot provide enough evidence for a positive correlation between the P2 promoter activity and expression levels of AgrBD. We are sorry about this and hope to get the reviewer’s understanding. We have replaced “Determination of AIP levels” with “Determination of the agr promoter P2 activity” in revised manuscript to ensure the rigorousness of this study. (Line 105 and 309)

5. Was a solvent control used in the biofilm formation assay in section 4.7?

Response: BHI with DMSO was used as the solvent control in biofilm formation assay. We have clarified this issue in section 4.7 in revised manuscript. (Line 331)

6. In line 307, the method of colony counting should be mentioned in the manuscript.

Response: We have mentioned the method of colony counting in revised manuscript. (Line 334-336)

7. Sections 4.10 and 4.12: Although the methodology has been previously described and the corresponding references have been provided in the manuscript, the author should still mention the methodology briefly (including essential information of the methods) in the manuscript

Response: We have described the method of FE-SEM analysis and hemolytic activity, adhesion and invasion assays in revised manuscript. (Line 364-369; 378-390)

8. I recommend moving Table A1 to the supplementary materials file.

Response: The reviewer’s comment is quite reasonable. We have moved Table A1 to the supplementary materials file.

Reviewer 2 Report

The manuscript entitled ‘Andrographolide Inhibits Biofilm and Virulence in Listeria monocytogenes as a Quorum Sensing Inhibitor ’ investigates the effect of andrographolide on two quorum sensing systems of L. monocytogenes. The results are interesting within the scope of Molecules. They are properly presented, and most of the methodological design is clear and appropriate. However, some remarks should be addressed before publication. My review comments:

Line 34 – Add a sentence or two about listeriosis infection

Line 64 – What is the relevance of the study? Please, rewrite the objective of the work highlighting the novelty of your research.

Line 71 – Why did you measure growth curves only up to the concentration of 1/2MIC? 1 mg/ml of andrographolide does not seem to influence the bacterial growth. How does it look like when the concentration of andrographolide is higher? The graph should present the growth curves of the cells incubated in the presence of andrographolide at a concentration up to 1 MIC.

Line 83 – The concentrations of andrographolide at ½ MIC and below shoud not be called “sub-inhibitory” because they do not affect bacterial growth.

Figure 3a – Why this figure shows decrease in OD595 at the increasing concentration of antibiotic while figure 1b shows that the increasing content of andrographolide did not affect the bacterial growth?

In addition, the manuscript lacks conclusion section summarizing the most important results of the study. Such section would be useful to the reader.

Author Response

Line 34 – Add a sentence or two about listeriosis infection

Response: The reviewer’s comment is quite reasonable. We have added sentences about listeriosis infection in revised manuscript. (Line 34-37)

Line 64 – What is the relevance of the study? Please, rewrite the objective of the work highlighting the novelty of your research.

Response: Thanks for the reviewer's kind suggestion. We have rewritten the objectives of this work in revised manuscript. (Line 69-76)

Line 71 – Why did you measure growth curves only up to the concentration of 1/2MIC? 1 mg/ml of andrographolide does not seem to influence the bacterial growth. How does it look like when the concentration of andrographolide is higher? The graph should present the growth curves of the cells incubated in the presence of andrographolide at a concentration up to 1 MIC.

Response: Thanks for the reviewer's kind suggestion. Growth curves were measured to determine the concentrations of andrographolide used for AI-2 and AIP production and biofilm formation assay. If the higher concentrations of andrographolide are used, growth of L. monocytogenes will be inhibited, leading to reduced AI-2 and AIP production and ability of biofilm formation. Therefore, appropriate concentrations of andrographolide should be determined to ensure that the difference in AI-2 and AIP production and biofilm formation ability is not due to a difference in bacterial growth. In general, concentrations of antimicrobial below the MIC value are selected for growth curves and investigating the effect of antimicrobial on signaling molecules production and biofilm formation ability. So growth curve in the presence of andrographolide at MIC was not measured in this study. We are sorry about this and hope to get the reviewer’s understanding.  

Line 83 – The concentrations of andrographolide at ½ MIC and below should not be called “sub-inhibitory” because they do not affect bacterial growth.

Response: Thanks for the reviewer's kind suggestion. We have changed “sub-inhibitory” to “sub-MICs”. (Line 83 and 94)

Figure 3a – Why this figure shows decrease in OD595 at the increasing concentration of antibiotic while figure 1b shows that the increasing content of andrographolide did not affect the bacterial growth?

Response: Figure 1B showed OD600nm values of the planktonic cells. While Figure 3A showed the biofilm biomass of L. monocytogenes. They were different. After incubation, biofilms were stained, decolorized and measured the absorbance at OD595nm. Andrographolide at 1/2MIC and below had no effect on growth of L. monocytogenes, but inhibited biofilm formation.

In addition, the manuscript lacks conclusion section summarizing the most important results of the study. Such section would be useful to the reader.

Response: Thanks for the reviewer's kind suggestion. We have added the conclusion section in revised manuscript. (Line 394-403)

Round 2

Reviewer 1 Report

The authors have satisfactorily addressed the issues raised by both reviewers. I would recommend the acceptance of the manuscript after the following points have been checked:

  1. There are a few grammatical mistakes in lines 229-237 (i.e. the paragraph added during revision). The authors should read this paragraph again carefully to fix these mistakes.

  1. While Table A1 has been deleted from the manuscript, it seems that it has not been included in the supplementary materials file. Would the authors please check and make sure that Table A1 is included in the supplementary materials file.

Author Response

1. There are a few grammatical mistakes in lines 229-237 (i.e. the paragraph added during revision). The authors should read this paragraph again carefully to fix these mistakes.

Response: Thanks for the reviewer's kind suggestion. We have corrected the grammatical mistakes.

2. While Table A1 has been deleted from the manuscript, it seems that it has not been included in the supplementary materials file. Would the authors please check and make sure that Table A1 is included in the supplementary materials file.

Response: We have placed Table A1 in supplementary material file and uploaded the new file. Maybe there is something wrong with our resubmission. We will upload the new version of supplementary material file.